# Using Shannon Entropy to Improve the Identification of MP-SBM Models with Undesirable Output

**DOI:** 10.3390/e24111608

**Published:** 2022-11-04

**Authors:** Zhanxin Ma, Jie Yin, Lin Yang, Yiming Li, Lei Zhang, Haodong Lv

**Affiliations:** 1School of Economics and Management, Inner Mongolia University, Hohhot 010021, China; 2School of Environment, Tsinghua University, Beijing 100084, China

**Keywords:** data envelopment analysis, MP-SBM, Shannon entropy model, public health events, efficiency assessment

## Abstract

In the context of the COVID-19 global epidemic, it is particularly important to use limited medical resources to improve the systemic control of infectious diseases. There is a situation where a shortage of medical resources and an uneven distribution of resources in China exist. Therefore, it is important to have an accurate understanding of the current status of the healthcare system in China and to improve the efficiency of their infectious disease control methods. In this study, the MP-SBM-Shannon entropy model (modified panel slacks-based measure Shannon entropy model) was proposed and applied to measure the disposal efficiency of the medical institutions responding to public health emergencies (disposal efficiency) in China from 2012 to 2018. First, a P-SBM (panel slacks-based measure) model, with undesirable outputs based on panel data, is given in this paper. This model measures the efficiency of all DMUs based on the same technical frontier and can be used for the dynamic efficiency analysis of panel data. Then, the MP-SBM model is applied to solve the specific efficiency paradox of the P-SBM model caused by the objective data structure. Finally, based on the MP-SBM model, undesirable outputs are considered in the original efficiency matrix alignment combination for the deficiencies of the existing Shannon entropy-DEA model. The comparative analysis shows that the MP-SBM-Shannon model not only solves the problem of the efficiency paradox of the P-SBM model but also improves the MP-SBM model identification ability and provides a complete ranking with certain advantages. The results of the study show that the disposal efficiency of the medical institutions responding to public health emergencies in China shows an upward trend, but the average combined efficiency is less than 0.47. Therefore, there is still much room for improvement in the efficiency of infectious disease prevention and control in China. It is found that the staffing problem within the Center for Disease Control and the health supervision office are two stumbling blocks.

## 1. Introduction

Data envelopment analysis (DEA) is a nonparametric method proposed by Charnes, Cooper, and Rhodes (CCR) to assess the relative effectiveness among DMUs [1]. The CCR model measures the production efficiency of a production system under constant returns to scale (CRS). The BCC model, given by Banker et al. [2] in 1984, is mainly used to evaluate the production efficiency under the variable returns to scale (VRS). On this basis, scholars have proposed many different DEA models for specific problems and have a wide range of applications in various performance evaluation problems [3,4,5]. However, both the CCR model and the BCC model are radial projection structures. In addition, in order to cope with Russell’s [6] non-commensurate measures of technical efficiency, Charnes et al. [7] introduced the additive or Pareto–Koopmans (PK) model, the first unit invariant slacks-based measure, which was later modified to some extent by Green [8]. Based on the above study, Tone [4] proposed a non-angle, non-radial SBM-DEA model in 2001, which does not need to consider the selection of orientation, and the inputs and outputs do not need to vary strictly proportionally. Later, Tone and Sahoo [9] constructed an undesirable output SBM-DEA model to address the impact of undesirable output on efficiency assessments. The model has been widely used in various fields, such as hospitals [10,11,12,13,14], banks [15,16,17,18,19], and educational institutions [20,21,22].

Due to the current reputation of COVID-19, the efficiency of healthcare institutions attracts great attention. Actually, some scholars have evaluated the efficiency of healthcare systems more in terms of their operational efficiency [23,24,25,26,27], technical efficiency [28,29,30], and resource allocation efficiency [31,32,33]. Additionally, some studies on infectious diseases have focused more on the prevention [34], control [35], and surveillance [36] of infectious diseases, while there are few studies about the healthcare systems in their responses to the public health emergency. In addition, there is a situation where there is a shortage of medical resources and an uneven distribution of resources in China. Therefore, it is important to have an accurate understanding of the current status of the healthcare system in China and to improve the efficiency of their infectious disease control methods.

In terms of the efficiency measurement of healthcare institutions, Sherman [37] was the first to use the DEA method, and afterwards, numerous scholars enriched this field of research [38,39,40,41]. In particular, the DEA method has also made great progress in evaluating the efficiency of Chinese healthcare organizations. For example, Liu [25] used the Super-SBM-DEA model to assess the efficiency of provincial healthcare services and their variation in China from 2008–2016. Zheng et al. [42] investigated a DEA method to assess the efficiency of the rural healthcare delivery system in 27 provinces in China from 2013 to 2017. Guo [43] applied a non-radially oriented distance function and a meta-boundary approach to analyze the regional healthcare efficiency in China from 2009 to 2019. Moreover, Campos et al. [44] found that the DEA method is an effective method for assessing the efficiency of healthcare systems. Karahan et al. [45] concluded that the DEA approach is an effective method of measuring hospital inefficiencies and can be used as a reference and basis to guide management, improve efficiency, and reduce healthcare costs. Due to the presence of undesirable output indicators, such as mortality in the process of infectious disease control, the SBM model [9] with undesirable outputs is particularly suitable for the efficiency evaluation among many DEA models and has been commonly employed in the field of healthcare efficiency evaluation [10,11,25]. Zhang et al. [11] held that the traditional DEA model overestimates the performance of health resource allocation in China, while the efficiency based on the SBM model with undesirable outputs is relatively lower than the traditional DEA model. Based on the above analysis, this paper applies the SBM method to measure the efficiency of the healthcare system in responding to public health emergencies.

However, the traditional SBM model has certain shortcomings. First, it does not allow an analysis of panel data, which is given based on sectional data. This model measures the efficiency of DMUs based on different technology frontier surfaces. Therefore, the efficiency values measured by this model are not comparable and cannot be compared longitudinally. Second, when using the panel data to construct a reference surface, the efficiency value will increase with the improvement of the reference standard, which is caused by the objective data structure [46,47,48]. Third, the traditional DEA model cannot fully sequence all DMUs, especially the more efficient ones. Fourth, they usually prefer the self-assessed weights to the evaluated units, which means that any advantages may be exaggerated and disadvantages may be ignored. Xie solved the third and fourth problems in 2014 by using Shannon entropy [49]. In this approach, they measured the efficiency matrix of all the subsets of indicators and used Shannon entropy to calculate the weights of each subset. The DEA-Shannon entropy method reduces extreme and unrealistic permutation weights by combining the efficiency values for all permutations as a matrix. Ma et al. [50] also argued that Shannon entropy makes the results more accurate and improves the identification ability of the DEA method. Scholars refer to the results measured by the DEA-Shannon entropy method as the comprehensive efficiency score (CES), and the method has been widely used to some extent due to its aforementioned advantages [50,51,52].

Nevertheless, this method still has some drawbacks. The method is based on the theory that a DEA model has at least one input and one output [53] to measure the efficiency matrix of all the subsets of variables. However, the permutations obtained above do not distinguish between the desirable and undesirable outputs and are not applicable to research problems with undesirable output indicators. For research with undesirable output indicators, the permutations will appear to have only a desirable or undesirable output. The absence of these outputs can bias the measurement results. Putting such permutations of efficiency values into the efficiency matrix will definitely lead to bias in the CES.

Based on the above analysis, the following work is conducted in this study. First, this paper provides a P-SBM model with undesirable outputs based on panel data. The model measures the technical efficiency of all the decision units based on the same technical frontier and, thus, can be used for the dynamic efficiency analysis of panel data. Second, this study proposes an MP-SBM model that can avoid the efficiency paradox in response to the specific efficiency paradox problem that arises in this study. Then, to address the shortcomings of the existing Shannon entropy-DEA model, the undesirable outputs are taken into account in the original efficiency matrix alignment combination. We rectify a deviation of the results of the original Shannon entropy-DEA model. Finally, this study constructs a new indicator system to evaluate the efficiency of healthcare systems. This indicator system takes into account the Center for Disease Control and Prevention (CDC), health surveillance agencies, and the Department of Infectious Disease (DID) and is more comprehensive than the original indicator system. Specifically, the input indicators include the health surveillance staff (WR), number of CDCs (JR), number of public health physicians (GY), total health expenditure (WF), and number of beds in DID (CC); the desirable output indicators include the number of outpatients in DID (CM) and the number of discharged patients in DID (CY); the undesirable output indicators include the number of patients in DID (CF) and the number of deaths in DID (CB). This study measures the disposal efficiency of 31 Chinese provinces from 2012 to 2018, which has positive implications for improving China’s public health service system and overall disposal efficiency.

The rest of this paper is organized as follows. Section 2 presents the P-SBM model and gives the MP-SBM model for the problems of the P-SBM model. Based on this, the MP-SBM-Shannon entropy model is given. Section 3 describes the indicator system and data sources. Section 4 analyzes the processing efficiency using the MP-SBM-Shannon entropy model and also compares the results with the MP-SBM and P-SBM models. Section 5 concludes the study. In summary, the research framework of this study is shown in Figure 1.

## 2. Methodology

### 2.1. SBM Model with Undesirable Outputs

To take the improvement of the slack variables into account, Tone [4] proposed an SBM model with undesirable outputs. This paper measures the disposal efficiency of 31 provinces in China from 2012 to 2018. Suppose there are *n* DMUs, each of which has *i* input indicators, *r* desirable output indicators, and *m* undesirable output indicators. Where $x_{ij}$ denotes the *i*-th input of the *j*-th DMU, $y_{rj}^{w}$ denotes the *r*-th desirable output of the *j*-th DMU, and $y_{mj}^{b}$ denotes the *m*-th undesirable output of the *j*-th DMU, denoted as $x_{ij}={\left(WR_{1j},JR_{2j},GY_{3j},WF_{4j},CC_{5j}\right)}^{T}>0$ , $y_{rj}^{w}={\left(CM_{1j}^{w},CY_{2j}^{w}\right)}^{T}>0$ , $y_{mj}^{b}={\left(CF_{1j}^{b},CB_{2j}^{b}\right)}^{T}>0$ , respectively. Then, the SBM model can be formulated as follows:
(1)(SBM)ρ=min1−15∑i=15si−xij01+14(∑r=12sr+yrj0w+∑m=12smbymj0b)s.t.∑j=1nWR1jλj+s1−=WR1j0,∑j=1nJR2jλj+s2−=JR2j0,∑j=1nGY3jλj+s3−=GY3j 0,∑j=1nWF4jλj+s4−=WF4j0,∑j=1nCC5jλj+s5−=CC5j0,∑j=1nCM1jwλj−s1+=CM1j0w,∑j=1nCY2jwλj−s2+=CY2j0w,∑j=1nCF1jbλj+s1b=CF1j0b,∑j=1nCB2jbλj+s2b=CB2j0b,δ(∑j=1nλj)=δ,λ=(λ1,⋯,λn)>=0,s−=(s1−,⋯,s5−)>=0,s+=(s1+,s2+)>=0,sb=(s1b,s2b)>=0.

where δ is a parameter that takes the value of 0 or 1; ρ is the efficiency of the *j*_0_-th DMU. When δ=0 , the production systems meet the constant returns to scale (CRS), ρ is the comprehensive efficiency value and measures the adequacy of resource utilization and the rationality of resource allocation in a provincial health system’s response to public health emergencies. When δ=1 , the production systems meet the variable returns to scale (VRS), ρ is the pure technical efficiency values and measures the technical level of each province’s healthcare system in the process of responding to public health emergencies. $s^{-}=(s_{1}^{-},\cdots ,s_{5}^{-})$ denotes the slack variables for the 5 inputs, $s^{+}=(s_{1}^{+},s_{2}^{+})$ denotes the slack variables for the 2 desirable outputs, $s^{b}=(s_{1}^{b},s_{2}^{b})$ denotes the slack variables for the 2 undesirable outputs, and $\lambda_{j}$ is the intensity variable associated with DMU*j*.

### 2.2. P-SBM Model with Undesirable Output for Measuring Panel Data

Since the traditional SBM model is based on sectional data, it cannot be analyzed for panel data. The efficiency values measured by this model are not comparable and cannot be compared longitudinally. Therefore, this paper gives a P-SBM model with an undesirable output based on panel data. The P-SBM model is based on the same technology frontier surface to measure the efficiency of all DMUs, has longitudinal comparability, and can be used for the dynamic efficiency analysis of panel data.

Suppose that a certain *n* DMU has *K* years of data for which the efficiency analysis is required, and the data in year *k*_0_ were selected as the reference surface. The input–output indicator for the *j*-th DMU in year *k* is $(WR_{1j}^{\left(k\right)},JR_{2j}^{\left(k\right)},GY_{3j}^{\left(k\right)},WF_{4j}^{\left(k\right)},$
$CC_{5j}^{\left(k\right)},CM_{1j}^{w(k)},CY_{2j}^{w(k)},CF_{1j}^{b(k)},CB_{2j}^{b(k)})>0$ , the efficiency value $\rho_{j,k_{0}}^{\left(k\right)}$ of the *j*-th DMU in year *k*, relative to the technology frontier in year *k*_0_, can be measured by models (2) and (3). If the DMU belongs to the reference set $T^{\left(k_{0}\right)}$ , then $\rho_{j,k_{0}}^{\left(k\right)}$ can be measured by model (2). If the DMU is inefficient, then $\rho_{j,k_{0}}^{\left(k\right)}$ must be less than 1. If the DMU is efficient, then $\rho_{j,k_{0}}^{\left(k\right)}$ must be equal to 1. If the DMU does not belong to the reference set $T^{\left(k_{0}\right)}$ , then there does not exist a production method in the reference set $T^{\left(k_{0}\right)}$ that is better than the DMU*j*. At this point, $\rho_{j,k_{0}}^{\left(k\right)}$ can be measured by model (3), and it must be greater than 1. When $\rho_{j,k_{0}}^{\left(k\right)}$ is larger, it means that the DMU is farther away from the frontier surface. The P-SBM model can be formulated as follows:


(2)
$$\begin{array}{c} \left\{\begin{array}{cc} \; & \psi_{1}=\min\frac{1-\frac{1}{5}\sum_{i=1}^{5}\frac{s_{i}^{-}}{x_{ij}^{\left(k\right)}}}{1+\frac{1}{4}(\sum_{r=1}^{2}\frac{s_{r}^{+}}{y_{rj}^{w(k)}}+\sum_{m=1}^{2}\frac{s_{m}^{b}}{y_{mj}^{b(k)}})}\\ \; & \begin{array}{cc} s.t. & \sum_{l=1}^{n^{\left(k_{0}\right)}}WR_{1l}^{\left(k_{0}\right)}\lambda_{l}+s_{1}^{-}=WR_{1j}^{\left(k\right)}{,}_{}\sum_{l=1}^{n^{\left(k_{0}\right)}}JR_{2l}^{\left(k_{0}\right)}\lambda_{l}+s_{2}^{-}=JR_{2j}^{\left(k\right)}, \end{array}\\ \; & \begin{array}{cc} & \begin{array}{cc} \; & \sum_{l=1}^{n^{\left(k_{0}\right)}}GY_{3l}^{\left(k_{0}\right)}\lambda_{l}+s_{3}^{-}=GY_{3j}^{\left(k\right)}{,}_{}\sum_{l=1}^{n^{\left(k_{0}\right)}}WF_{4l}^{\left(k_{0}\right)}\lambda_{l}+s_{4}^{-}=WF_{4j}^{\left(k\right)},\\ \; & \sum_{l=1}^{n^{\left(k_{0}\right)}}CC_{5l}^{\left(k_{0}\right)}\lambda_{l}+s_{5}^{-}=CC_{5j}^{\left(k\right)}{,}_{}\sum_{l=1}^{n^{\left(k_{0}\right)}}CM_{1l}^{w(k_{0})}\lambda_{l}-s_{1}^{+}=CM_{1j}^{w(k)},\\ \; & \sum_{l=1}^{n^{\left(k_{0}\right)}}CY_{2l}^{w(k_{0})}\lambda_{l}-s_{2}^{+}=CY_{2j}^{w(k)}{,}_{}\sum_{l=1}^{n^{\left(k_{0}\right)}}CF_{1l}^{b(k_{0})}\lambda_{l}+s_{1}^{b}=CF_{1j}^{b(k)},\\ \; & \sum_{l=1}^{n^{\left(k_{0}\right)}}CB_{2l}^{b(k_{0})}\lambda_{l}+s_{2}^{b}=CB_{2j}^{b(k)}{,}_{}\delta \left(\sum_{l=1}^{n^{\left(k_{0}\right)}}\lambda_{l}\right)=\delta , \end{array} \end{array}\\ & \begin{array}{cc} & \begin{array}{cc} \; & \;\lambda =\left(\lambda_{1},\cdots ,\lambda_{n}\right)\underset{=}{>}0,s^{-}=\left(s_{1}^{-},\cdots ,s_{5}^{-}\right)\underset{=}{>}0,\\ \; & s^{+}=\left(s_{1}^{+},s_{2}^{+}\right)\underset{=}{>}0,s^{b}=\left(s_{1}^{b},s_{2}^{b}\right)\underset{=}{>}0. \end{array} \end{array} \end{array}\right. \end{array}$$



(3)
$$\begin{array}{c} \left\{\begin{array}{cc} \; & \psi_{2}=\min\frac{1+\frac{1}{5}\sum_{i=1}^{5}\frac{s_{i}^{-}}{x_{ij}^{\left(k\right)}}}{1-\frac{1}{4}(\sum_{r=1}^{2}\frac{s_{r}^{+}}{y_{rj}^{w(k)}}+\sum_{m=1}^{2}\frac{s_{m}^{b}}{y_{mj}^{b(k)}})}\\ \; & \begin{array}{cc} s.t. & \sum_{l=1}^{n^{\left(k_{0}\right)}}WR_{1l}^{\left(k_{0}\right)}\lambda_{l}-s_{1}^{-}\underset{=}{<}WR_{1j}^{\left(k\right)}{,}_{}\sum_{l=1}^{n^{\left(k_{0}\right)}}JR_{2l}^{\left(k_{0}\right)}\lambda_{l}-s_{2}^{-}\underset{=}{<}JR_{2j}^{\left(k\right)}, \end{array}\\ \; & \begin{array}{cc} & \begin{array}{cc} \; & \sum_{l=1}^{n^{\left(k_{0}\right)}}GY_{3l}^{\left(k_{0}\right)}\lambda_{l}-s_{3}^{-}\underset{=}{<}GY_{3j}^{\left(k\right)}{,}_{}\sum_{l=1}^{n^{\left(k_{0}\right)}}WF_{4l}^{\left(k_{0}\right)}\lambda_{l}-s_{4}^{-}\underset{=}{<}WF_{4j}^{\left(k\right)},\\ \; & \sum_{l=1}^{n^{\left(k_{0}\right)}}CC_{5l}^{\left(k_{0}\right)}\lambda_{l}-s_{5}^{-}\underset{=}{<}CC_{5j}^{\left(k\right)}{,}_{}\sum_{l=1}^{n^{\left(k_{0}\right)}}CM_{1l}^{w(k_{0})}\lambda_{l}+s_{1}^{+}\underset{=}{>}CM_{1j}^{w(k)},\\ \; & \sum_{l=1}^{n^{\left(k_{0}\right)}}CY_{2l}^{w(k_{0})}\lambda_{l}+s_{2}^{+}\underset{=}{>}CY_{2j}^{w(k)}{,}_{}\sum_{l=1}^{n^{\left(k_{0}\right)}}CF_{1l}^{b(k_{0})}\lambda_{l}-s_{1}^{b}\underset{=}{<}CF_{1j}^{b(k)},\\ \; & \sum_{l=1}^{n^{\left(k_{0}\right)}}CB_{2l}^{b(k_{0})}\lambda_{l}-s_{2}^{b}\underset{=}{<}CB_{2j}^{b(k)}{,}_{}\delta \left(\sum_{l=1}^{n^{\left(k_{0}\right)}}\lambda_{l}\right)=\delta ,s^{+}\underset{=}{<}y_{j}^{w(k)}, \end{array} \end{array}\\ & \begin{array}{cc} & \begin{array}{cc} \; & \;\lambda =\left(\lambda_{1},\cdots ,\lambda_{n}\right)\underset{=}{>}0,s^{-}=\left(s_{1}^{-},\cdots ,s_{5}^{-}\right)\underset{=}{>}0,\\ \; & s^{+}=\left(s_{1}^{+},s_{2}^{+}\right)\underset{=}{>}0,s^{b}=\left(s_{1}^{b},s_{2}^{b}\right)\underset{=}{>}0. \end{array} \end{array} \end{array}\right. \end{array}$$


The production possibility set for year *k*_0_ is shown as follows:(4)$$T^{\left(k_{0}\right)}=\left\{(x,{\mathrm{>y}}^{w},y^{b})\left|x\underset{=}{>}\sum_{l=1}^{n}x_{l}^{\left(k_{0}\right)}\lambda_{l},y^{w}\underset{=}{<}\sum_{l=1}^{n}y_{l}^{\left(k_{0}\right)}\lambda_{l},y^{b}\underset{=}{>}\sum_{l=1}^{n}y_{l}^{\left(k_{0}\right)}\lambda_{l},\delta \sum_{l=1}^{n}\lambda_{l}=\delta ,\lambda =(\lambda_{1},\cdots ,\lambda_{n})\underset{=}{>}0\right.\right\}$$

Assuming that the production system satisfies the variable returns to scale, the set of production possibilities $T^{\left(k_{0}\right)}$ determined by DMU *A* and DMU *B* is shown in the shaded portion of Figure 2.

For DMU_1_, if DMU *D* is taken as its improvement target, $s_{1}^{-},s_{1}^{+}$ represents the deficiencies of DMU_1_ in terms of the input and output indicators, respectively. To eliminate the influence of the indicator magnitude on the evaluation results, $s_{1}^{-},s_{1}^{+}$ is divided by $x_{1},y_{1}^{}$ , respectively, where $s_{1}^{-}/x_{1}$ indicates the ratio of the shortfall of the estimated unit’s inputs relative to the amount of inputs and $s_{1}^{+}/y_{1}^{}$ denotes the percentage of the shortfall in the output of the evaluated unit relative to the amount of output. The degree of effectiveness of DMU_1_ relative to DMU *D* can be expressed by the following equation:(5)$$\rho_{1}=\frac{1-s_{1}^{-}/x_{1}}{1+s_{1}^{+}/y_{1}^{}}$$

For DMU_2_, if its effective reference point is chosen as point *C*, $s_{2}^{-},s_{2}^{+}$ represents the extent to which it outperforms the effective reference point *C* in terms of the input and output indicators, respectively. Similarly, the degree of effectiveness of DMU_2_ for DMU *C* can be expressed by the following equation:(6)$$\rho_{2}=\frac{1+s_{2}^{-}/x_{2}}{1-s_{2}^{+}/y_{2}^{}}$$

Since the efficiency measures for all the above DMUs are based on the same technological frontier, the efficiency values given by models (2) and (3) can be used for the analysis of dynamic efficiency changes in the panel data.

### 2.3. Problems and Causes of the P-SBM Model

However, there is an efficiency paradox when evaluating the efficiency of public health systems based on the P-SBM model. Specifically, when using panel data to construct a reference surface, the measurements will show that the higher the reference standard, the greater the efficiency value caused by the objective data structure. The following section illustrates the possible efficiency paradox of the P-SBM model through an example [46].

#### 2.3.1. Quantitative Analysis of Disposal Efficiency in Tibet from 2012 to 2017

First, to make the technological progress changes more prominent, the time span of constructing the evaluation reference set is widened. The production frontier of the Chinese provinces in 2012 and 2017 is selected as the evaluation reference set to estimate the infectious disease disposal efficiency of medical institutions in Tibet. Figure 3 gives the pure technical efficiency status of Tibet from 2012 to 2017. Among them, the number of public health-practicing (assistant) physicians in the medical institutions in each region was selected as the input indicator, the number of infectious disease discharges in each region was selected as the output indicator, and the results were obtained by applying the P-SBM model as follows: $\rho_{\mathrm{Tibet},2012}^{\left(k\right)}{}_{}{<}_{}\rho_{\mathrm{Tibet},2017}^{\left(k\right)},$  *k* = 2012, 2013, 2014.

#### 2.3.2. Qualitative Analysis Results

China has prioritized infectious disease public health events and has been increasing financial investment and policy support for them [54,55]. Meanwhile, with the continuous advancement of medical technology in recent years, China’s medical level has comprehensively improved [56,57]. This was confirmed by a related study in the international medical journal, The Lancet, which concluded that China is one of the countries with the greatest progress in healthcare quality in the last 25 years.

Therefore, from the perspective of technological progress, the efficiency of infectious disease disposal in Tibet from 2012 to 2017 is estimated using the production frontier of the Chinese provinces in 2012 and 2017 as the evaluation reference system. The obtained efficiency values should decrease with the increase in the technological level of the evaluation reference system, namely:$$\rho_{\mathrm{Tibet},2012}^{\left(k\right)}\geq \rho_{\mathrm{Tibet},2017}^{\left(k\right)}{,}_{}k=2012,\;2013,\;2014,$$
This shows that the results given by the P-SBM model contradict the actual situation.

#### 2.3.3. Analysis of the Causes of the Efficiency Paradox

The following data analysis reveals that the reason for the efficiency paradox in the P-SBM model is the data’s short-tail phenomenon. Figure 4 provides the effective frontier for physicians curing inpatients in 2012 and 2017 for each province in China. Among them, the production frontier composed of data in 2012 is line *CDF*, and the production frontier composed of data in 2017 is line *ABE*. Point *G* (454, 2848) is Tibet’s input and output data for 2013.

Since point *G* is in the evaluation reference set $T^{\left(2012\right)}$ , we have $\rho_{\mathrm{Tibet},2012}^{(2013)}\underset{=}{<}1$ . Additionally, since point *G* is not in the evaluation reference set $T^{\left(2017\right)}$ , we obtain $\rho_{\mathrm{Tibet},2017}^{(2013)}>1$ . Hence, we obtain that $\rho_{\mathrm{Tibet},2012}^{(2013)}<\rho_{\mathrm{Tibet},2017}^{(2013)}$ . The reason for this efficiency paradox is the lack of data in $T^{\left(2017\right)}$ . Since the lowest number of public health physicians in all the provinces in 2017 is Qinghai, with 496, corresponding to point *A* in Figure 4 it leads to the absence of input data with less than 454 in $T^{\left(2017\right)}$ , thus $G\notin T^{\left(2017\right)}$ , i.e., $\rho_{\mathrm{Tibet},2017}^{(2013)}>1$ . In fact, the input–output value at point *C* (Tibet, 2012) is also achievable in 2017 because the technological progress is irreversible. Therefore, point *C* should also belong to $T^{\left(2017\right)}$ . If $C\in T^{\left(2017\right)}$ , then we have $G\in T^{\left(2017\right)}$ and, hence, $\rho_{\mathrm{Tibet},2017}^{(2013)}\underset{=}{<}1$ . The above analysis reveals that there may be an efficiency paradox in the results given by the P-SBM model. Therefore, we applied the MP-SBM model to measure the disposal efficiency in this paper.

### 2.4. MP-SBM Model

Suppose there are *K* years of data to be analyzed for efficiency, where there are $n^{\left(k\right)}$ DMUs in year *k*, and the value of the input–output indicator for the *j*-th DMU in year *k* is $(WR_{j}^{\left(k\right)},JR_{j}^{\left(k\right)},GY_{j}^{\left(k\right)},WF_{j}^{\left(k\right)},CC_{j}^{\left(k\right)},CM_{j}^{w(k)},CY_{j}^{w(k)},CF_{j}^{b(k)},CB_{j}^{b(k)})>0$ . In the case of irreversible technological progress, since the input–output situation before year *K* can also be realized in year *K*, the evaluation reference set $T^{\left(K\right)}$ for year *K* can be expressed as follows:(7)$$T^{\left(K\right)}=\{(x,y^{w},y^{b})|x\underset{=}{>}\sum_{t=1}^{K}\sum_{j=1}^{n^{\left(t\right)}}x_{j}^{\left(t\right)}\lambda_{j}^{\left(t\right)},y^{w}\underset{=}{<}\sum_{t=1}^{K}\sum_{j=1}^{n^{\left(t\right)}}y_{j}^{w(t)}\lambda_{j}^{\left(t\right)},\;y^{b}\underset{=}{>}\sum_{t=1}^{K}\sum_{j=1}^{n^{\left(t\right)}}y_{j}^{b(t)}\lambda_{j}^{\left(t\right)},\delta \sum_{t=1}^{K}\sum_{j=1}^{n^{\left(t\right)}}\lambda_{j}^{\left(t\right)}=\delta ,\lambda_{j}^{\left(t\right)}\underset{=}{>}0,j=1,2,\ldots ,n^{\left(t\right)},t=1,2,\ldots ,K\},$$
Therefore, for the *p*-th DMU in year *k*, the MP-SBM model can be obtained as follows:
(8)$$\begin{array}{c} \left\{\begin{array}{cc} \; & \rho_{p}^{\left(k\right)}=\min\frac{1-\frac{1}{5}\sum_{i=1}^{5}\frac{s_{i}^{-}}{x_{ip}^{\left(k\right)}}}{1+\frac{1}{4}(\sum_{r=1}^{2}\frac{s_{r}^{+}}{y_{rp}^{w(k)}}+\sum_{m=1}^{2}\frac{s_{m}^{b}}{y_{mp}^{b(k)}})}\\ \; & \begin{array}{cc} s.t. & \sum_{t=1}^{K}\sum_{j=1}^{n^{\left(t\right)}}WR_{1j}^{\left(t\right)}\lambda_{j}^{\left(t\right)}+s_{1}^{-}=WR_{1p}^{\left(k\right)},\sum_{t=1}^{K}\sum_{j=1}^{n^{\left(t\right)}}JR_{2j}^{\left(t\right)}\lambda_{j}^{\left(t\right)}+s_{2}^{-}=JR_{2p}^{\left(k\right)}, \end{array}\\ \; & \begin{array}{cc} & \begin{array}{cc} \; & \sum_{t=1}^{K}\sum_{j=1}^{n^{\left(t\right)}}GY_{3j}^{\left(t\right)}\lambda_{j}^{\left(t\right)}+s_{3}^{-}=GY_{3p}^{\left(k\right)},\sum_{t=1}^{K}\sum_{j=1}^{n^{\left(t\right)}}WF_{4j}^{\left(t\right)}\lambda_{j}^{\left(t\right)}+s_{4}^{-}=WF_{4p}^{\left(k\right)},\\ \; & \sum_{t=1}^{K}\sum_{j=1}^{n^{\left(t\right)}}CC_{5j}^{\left(t\right)}\lambda_{j}^{\left(t\right)}+s_{5}^{-}=CC_{5p}^{\left(k\right)},\sum_{t=1}^{K}\sum_{j=1}^{n^{\left(t\right)}}CM_{1j}^{w(t)}\lambda_{j}^{\left(t\right)}-s_{1}^{+}=CM_{1p}^{w(k)},\\ \; & \sum_{t=1}^{K}\sum_{j=1}^{n^{\left(t\right)}}CY_{2j}^{w(l)}\lambda_{j}^{\left(l\right)}-s_{2}^{+}=CY_{2p}^{w(k)},\sum_{t=1}^{K}\sum_{j=1}^{n^{\left(t\right)}}CF_{1j}^{b(t)}\lambda_{j}^{\left(t\right)}+s_{1}^{b}=CF_{1p}^{b(k)} \end{array} \end{array}\\ & \begin{array}{cc} & \;\sum_{t=1}^{K}\sum_{j=1}^{n^{\left(t\right)}}CB_{2j}^{b(t)}\lambda_{j}^{\left(t\right)}+s_{2}^{b}=CB_{2p}^{b(k)},\delta \left(\sum_{t=1}^{K}\sum_{j=1}^{n^{\left(t\right)}}\lambda_{j}^{\left(t\right)}\right)=\delta \end{array}\\ & \begin{array}{cc} & \;s^{-}\underset{=}{>}0,s^{+}\underset{=}{>}0,s^{b}\underset{=}{>}0,\lambda_{j}^{\left(l\right)}\underset{=}{>}0,j=1,2,\cdots ,n^{\left(t\right)},t=1,2,\cdots ,K \end{array} \end{array}\right. \end{array}$$
Here, $\rho_{p}^{\left(k\right)}$ is the efficiency value of the *p*-th DMU in the year *k*. When δ=0 , the production system satisfies the constant returns to scale (CRS); when δ=1 , the production system meets the variable returns to scale (VRS).

If the optimal solution of model (6) is $\lambda_{j}^{\left(t\right)},j=1,2,\ldots ,n^{\left(t\right)},t=1,2,\ldots ,K,s^{-0},s^{+0},s^{b0},$ let
(9)$${\hat{x}}_{p}^{\left(k\right)}=x_{p}^{\left(k\right)}-s^{-0},{\hat{y}}_{p}^{w(k)}=y_{p}^{w(k)}+s^{+0},{\hat{y}}_{p}^{b(k)}=y_{p}^{b(k)}-s^{b0}$$
and call $\left({\hat{x}}_{p}^{\left(k\right)},{\hat{y}}_{p}^{w(k)},{\hat{y}}_{p}^{b(k)}\right)$ as the projection of the DMU*p*.

### 2.5. MP-SBM-Shannon Entropy Model

The concept of entropy originated in physics as a measurement of the degree of disorder in a thermodynamic system. Within information theory, entropy is a measurement of uncertainty. We know that the normal state of the particle is “disorderly motion”, and “entropy” can be considered as a measurement of “disorderly”. In a system, the greater the entropy, the greater the disorder and the smaller the corresponding weight. Thus, through entropy, we can reduce the role of the unordered part and amplify the role of the ordered part. In 1948, Shannon [58] introduced Shannon entropy, which is defined as the probability of occurrence of discrete random events. Therefore, Shannon entropy is used to describe the degree of information uncertainty.

Corresponding to the probability distribution P = (*p*_1_, ……, *p_n_*), (here, 0 ≤ *p_i_* ≤ 1 (*i* = 1, 2, …, *n*), and $\sum_{i=1}^{n}p_{i}=1$), and the Shannon entropy can be described as $H(p)=-\sum_{i=1}^{n}p_{i}\log p_{i}$ . When the degree of uncertainty of the probability distribution *P* is greater, its corresponding entropy *H* (P) value is larger; conversely, its entropy *H* (P) value is smaller.

To the best of our knowledge, the research on combining Shannon entropy and the DEA method is divided into two main areas. The first one uses Shannon entropy to integrate the different DEA model measurements [37,38]. The second one obtains the CES by evaluating the importance of each combination of variables to improve the DEA identification and provides a complete ranking [11,39]. In this paper, the original ranking combination is improved somewhat by taking into account the undesirable output indicators based on the study conducted by Xie [49].

Since the efficiency matrix, derived from the original DEA-Shannon entropy model, does not distinguish between the desired and undesired outputs, there may be cases where the measurement results deviate from the problem under study or are meaningless. For example, when we measure the efficiency of inter-provincial green development, we need to consider energy and resource consumption, economic growth, and environmental pollution emissions at the same time. When no distinction is made between the desired and undesired outputs, a permutation of energy consumption, economic growth, energy consumption, and environmental pollution emissions occurs. Additionally, the efficiency measured under this permutation will no longer be the green development efficiency, thus biasing the measured CES results.

Therefore, this paper needs to ensure that each portfolio of indicators contains at least one input, one desirable output, and one undesirable output. The number of all different combinations of input subsets from *M*, desirable output subsets from *S*, and undesirable output subsets from *T* is $K=(2^{m}-1)(2^{s}-1)(2^{t}-1)=(2^{5}-1)(2^{2}-1)(2^{2}-1)=279$ . Denote the *k*-th combination of the variable set as *M_k_*, and the model set is $\Omega =\left\{M_{1}{,M}_{2},\ldots {,M}_{K}\right\}$ . We denote the efficiency score of DMU*_j_* based on *M_k_* as *E_jk_*, j=1,2,…,n,k=1,2,…,K . There are 217 DMUs (31 provinces in 2012–2018), and we used MATLAB to calculate the efficiency matrix of 217 × 279, which yields [*E_jk_*]*_n×K_* as follows:(10)$$\begin{array}{l} \begin{array}{ccccc} \; & M_{1} & M_{2} & \ldots & M_{K} \end{array}\\ \begin{array}{c} {\mathrm{DMU}}_{1}\\ {\mathrm{DMU}}_{2}\\ \vdots \\ {\mathrm{DMU}}_{n} \end{array}\left[\begin{array}{cccc} E_{11} & E_{12} & \ldots & E_{1K}\\ E_{21} & E_{22} & \ldots & E_{2K}\\ \vdots & \vdots & \ddots & \vdots \\ E_{n1} & E_{n2} & \ldots & E_{nK} \end{array}\right] \end{array}$$

After measuring the efficiency matrix, the calculation process is as follows:
Step 1: Normalize the efficiency Matrix [*E_jk_*]*_n×K_* and set $e_{jk}=E_{jk}/\sum_{j=1}^{n}E_{jk},k=1,2,\ldots ,K.$Step 2: Compute entropy *f_k_* as $f_{k}=-{\left(\ln n\right)}^{-1}\sum_{j=1}^{n}e_{jk}\ln(e_{jk}),k=1,2,\ldots ,K.$Step 3: Calculate the degree of the diversification of *M_k_* as $d_{k}=1-f_{k},k=1,2,\ldots ,K.$Step 4: Normalize the value of *d_k_* as $W_{k}=d_{k}/\sum_{k=1}^{K}d_{k},k=1,2,\ldots ,K,$ such that $\sum_{k=1}^{K}W_{k}=1.$Step 5: Calculate the CES as $\theta_{j}=\sum_{k=1}^{K}W_{k}E_{jk},j=1,2,\ldots ,n.$

## 3. Indicator System and Data Sources

### 3.1. Selection of Indicator System

The indicator system is crucial to reflect the efficiency of the DMU accurately. Based on the indicators used by scholars (as shown in Table 1), this paper comprehensively considers the detection, control and treatment of infectious diseases. Compared with the original index system, this index system increases the number of health surveillance staff and the number of CDCs as human input indicators, which can better reflect the role played by the health supervision offices and CDCs in the detection and control process of infectious diseases. Additionally, this study considers undesirable output indicators, which is a more careful consideration.

Based on the above studies, this paper establishes a new evaluation indicator system, as shown in Table 2.

### 3.2. Data Sources and Descriptive Statistical Analysis

The data relating to the healthcare system of each province in China are collected according to the indicator system in Table 2. All indicator data are obtained from the China Health Care Statistics Yearbook (2012–2019) and the China Population and Employment Statistics Yearbook (2012–2019). The descriptive statistical results of the relevant data are shown in Table 3.

As can be seen from Table 3, the average total health expenditure, the number of beds in the infectious disease department, and the number of outpatients and discharged patients in the DID discharges grew by 108.4%, 22.8%, 64.0%, and 27.0%, respectively. It illustrates that the financial and material investment in medical institutions and public health institutions related to infectious diseases prevention and treatment has increased, and the medical institutions’ capacity has improved. However, the number of infectious disease cases did not drop significantly, with the number of deaths increasing by 18.4%, indicating that the control of infectious diseases remains a difficult task. The significant extreme differences in each indicator show that the provinces are still very uneven regarding infectious disease resource allocation and control capacity.

## 4. Results and Discussion

### 4.1. Comparison of Disposal Efficiency Based on MP-SBM-Shannon Entropy Model, MP-SBM Model, and P-SBM Model

To make the changes in technological progress more evident through the measurement results based on the P-SBM model, the data of the Chinese provinces in 2012 and 2018 are selected as the reference surface for evaluation, respectively. Additionally, the MP-SBM model uses the data of all Chinese provinces from 2012 to 2018 as the reference surface. The MP-SBM-Shannon model measured the efficiency matrix of 217 × 279 based on the MP-SBM model and measured the CES. Due to space limitation, the different combinations of variables with the top 15 weight values are selected and listed in Table 4, where “0” and “1” denote “not selected” and “selected” for the indicator, respectively.

Figure 5 presents the disposal efficiency based on the MP-SBM model, P-SBM model, and MP-SBM-Shannon entropy model, respectively. It can be seen that the three models are ranked in their descending order of efficiency values as the P-SBM model (2012), P-SBM model (2018), MP-SBM model, and the MP-SBM-Shannon entropy model, with efficiency values of 0.80, 0.69, 0.57, and 0.42, respectively, with a large difference. Additionally, it can be observed that there is a significant difference in the ranking of provinces on disposal efficiency based on the different models (as shown in Table 5). The MP-SBM model avoids the efficiency paradox that exists in the P-SBM model. However, the MP-SBM model has several provinces with efficiency values of 1, which cannot be ranked. The MP-SBM-Shannon entropy model not only provides detailed rankings but also considers 279 different permutations, which makes the measured results more reasonable.

### 4.2. Spatial and Temporal Analyses of the Disposal Efficiency

Figure 6 reports the average trend of disposal efficiency in China from 2012 to 2018 based on the MP-SBM-Shannon entropy model.

As shown in Figure 6, the disposal efficiency showed an upward trend, although it fluctuated slightly. This indicates that the Chinese medical and public health institutions’ efficiency in preventing and treating infectious diseases is increasing. It results from high national attention and the continuous advancement of medical reform. However, the disposal efficiency ranged from 0.38 to 0.47 during the study period, which indicates that the healthcare systems have a low resource allocation capacity in response to public health emergencies. The average disposal efficiency values increased to some extent in 2014, with efficiency values of 0.43. Combining the disposal efficiency trends by region, this is mainly caused by the efficiency changes in the central and northeastern regions. This is mainly due to the enormous efficiency gains in Hubei and Hunan in the central region and Liaoning in the northeast region in 2014. The input–output data found that the number of outpatients and discharged patients increased in these three provinces in 2014, while the number of patients and deaths in DID decreased. As a result, the disposal efficiency improved in 2014 compared to 2013.

Figure 7 illustrates the regional disposal efficiency and growth rate from 2012 to 2018 based on two models. In terms of the average disposal efficiency, it can be seen that the eastern region gained the highest efficiency, while the northeastern region gained the lowest efficiency. Additionally, in terms of the growth rate, the faster growth rate among the four economic regions occurred in the eastern and western regions.

Figure 8 gives the provincial disposal efficiency from 2012 to 2018. As shown in Figure 7 and Figure 8, (i) the efficiency in the eastern region is higher than in the other regions and presents a clear uptrend, and the average disposal efficiency is 0.50, with a growth rate of 32.6%. This is mainly attributed to Jiangsu, Zhejiang, and Beijing, whose average disposal efficiencies are more significant than 0.7. This reflects the generally higher level of medical care in the eastern region, which is also more efficient in managing infectious diseases than other regions. The worst-performing province was Hainan, with an average disposal efficiency value of 0.25, and the reasons for the lower efficiency need to be further identified. (ii) The growth rate of the average efficiency in the central region is negative, although its efficiency level is relatively high. The central region’s average disposal efficiency is 0.42, but the growth rate is −3.6%. Among them, Anhui has an average disposal efficiency value of 0.71, while Shanxi has the worst performance, with an average efficiency value of only 0.19. (iii) The efficiency of the western region is at a moderately low level, but its efficiency growth rate is at a high level. The average disposal efficiency is 0.38, and the growth rate is 47.7%. Gansu, Inner Mongolia, has a low disposal efficiency, with efficiency values below 0.28. (iv) The efficiency level and growth rate in the northeast region are low; the disposal efficiency is 0.35, and the growth rate is −21.43%. This indicates an urgent need for the northeast to analyze the region’s efficiency deficiencies further and accelerate improvements in the efficiencies of medical and public health institutions in dealing with infectious diseases.

Overall, the MP-SBM-Shannon model improves the identification ability of the MP-SBM model. The efficiency values do not vary much between regions, but there are considerable differences across the provinces. It can be seen, however, that high disposal efficiency scores may come from economically developed coastal provinces (Jiangsu, Zhejiang, etc.) or the less developed western regions (Tibet, Xinjiang, etc.). Since the reform and opening up, the coastal provinces have received rapid development, are richer in medical resources, and have actively adopted advanced technology, equipment, and management experience, which may account for the higher disposal efficiency in the more economically developed coastal provinces. In recent years, the country has increased its efforts to reform the western region, which has led to a significant increase in disposal efficiency in the region; therefore, low levels of economic development are not necessarily associated with low disposal efficiency. From the above analysis, we know that the overall efficiency level of China’s healthcare system has dramatically improved. However, there is still much room for improvement in the efficiency of infectious disease prevention and control, and some of the less efficient provinces should be further analyzed.

### 4.3. Analysis of Inefficient Provinces

From the above analysis, it can be seen that five provinces, Hainan, Shanxi, Gansu, Inner Mongolia, and Heilongjiang, have relatively low efficiencies, with an average score of no more than 0.29 during the study period. Therefore, further analysis of the efficiency status of these five provinces is given below, and the relevant results are shown in Figure 9, Figure 10 and Figure 11.

Figure 9 provides the average proportion of each input factor that can be improved. A larger ratio indicates a greater redundancy for that input indicator. It can be seen that the five inefficient regions have large redundancies and poor resource optimizations in various input indicators. The CDC (JR) and health supervision (WR) staffing problems are more prominent, in which the redundancy of the number of CDCs exceeds 60% in all regions. Therefore, the efficiency of staff needs to be further improved. In addition, in terms of the number of public health physicians (GY) and infectious disease beds (CC), both Shanxi and Inner Mongolia have redundancy levels above 49%, indicating that both regions are optimized above 49% to be effective. Some redundancy also exists in Hainan regarding the number of public health physicians, and Heilongjiang and Gansu, in terms of the utilization of total health costs. Therefore, further measures should be taken to optimize the allocation of resources, improve the quality and efficiency of staff, and improve the overall levels of prevention and control of infectious disease public health events in these areas.

Second, in terms of the desirable outputs, Figure 10 provides the average ratio of improvements that could be made for each desirable output indicator. A larger ratio indicates a greater degree of deficiency for that output indicator. As seen in Figure 10, Hainan has a significant deficit in the number of outpatients in DID (CM), which illustrates its inadequate resource utilization and service efficiency. In addition, Shanxi, Heilongjiang, and Gansu also have some room for improvement in outpatient attendance. Therefore, the regions should further improve the capacity and level of the outpatient services.

Finally, from the perspective of undesirable output, Figure 11 provides the proportion of each undesirable output indicator that can be improved, with a smaller ratio indicating that the region has better control over the undesired output. Figure 11 shows that all regions have some deficiencies in controlling the number of infectious disease incidents (CF) and infectious disease deaths (CB). Among them, Hainan, Shanxi, and Inner Mongolia need effective control in terms of the number of infectious disease incidences, with an improvement of 60% or more to reach the effective degree. From the above analysis, we can discover the shortcomings of these provinces to put forward constructive suggestions for following regulations.

## 5. Conclusions

Specifically, we proposed an MP-SBM model to avoid the efficiency paradox of the P-SBM model, and we considered the efficiency matrix with undesirable outputs in the MP-SBM-Shannon model. The efficiency gap between the three models is large, and there is a significant change in the ranking of disposal efficiency in all provinces. Through the analysis, the P-SBM model has an efficiency paradox, and the MP-SBM model has more reasonable and accurate results. The MP-SBM-Shannon model improves the identification ability of the MP-SBM model. In addition, this paper establishes a more suitable indicator system for evaluating disposal efficiency.

This paper applied the MP-SBM-Shannon model to measure the efficiency of China’s healthcare system in responding to public health emergencies from 2012 to 2018. We found that improving the staffing efficiency of CDCs and health supervision offices should be prioritized, and the efficiency of outpatient treatment also needs to be improved. Disposal efficiency varies widely among regions, and the reasons for the growth in disposal efficiency vary across regions. The results show that the disposal efficiency showed an overall increase during the study period; however, the efficiency values varied widely among provinces. Specifically, Anhui, Jiangsu, and Zhejiang have the highest disposal efficiency, while Hainan, Inner Mongolia, and Shanxi performed worst. Overall, the eastern region performed well in terms of efficiency values and growth trends, while the central and northeastern regions showed negative growth trends in efficiency values. The poorly performing provinces can be considered an enormous redundancy in various inputs, with a lack of capacity for optimal resource allocation.

Based on the results obtained from our study, we propose some suggestions as follows:
(1)In the long run, the staffing efficiency of CDCs and health supervision offices should be prioritized, and the efficiency of outpatient treatment also needs to be improved. The analysis of the inefficient provinces found that the staffing problem of the CDC and the health supervision office are two stumbling blocks. We need to develop an emergency management system to reduce personnel redundancy during normal times while alleviating the shortage of medical resources in the face of public health emergencies.(2)Disposal efficiency varies widely among these regions, which can be attributed to the differentiating growth in disposal efficiency. Therefore, paying attention to the optimal allocation of resources and improving the overall distribution of resources is urgently required. Additionally, it should accelerate technological progress and focus on improving management capabilities rather than blind investment and low-level expansion.

## Figures and Tables

**Figure 1 entropy-24-01608-f001:**
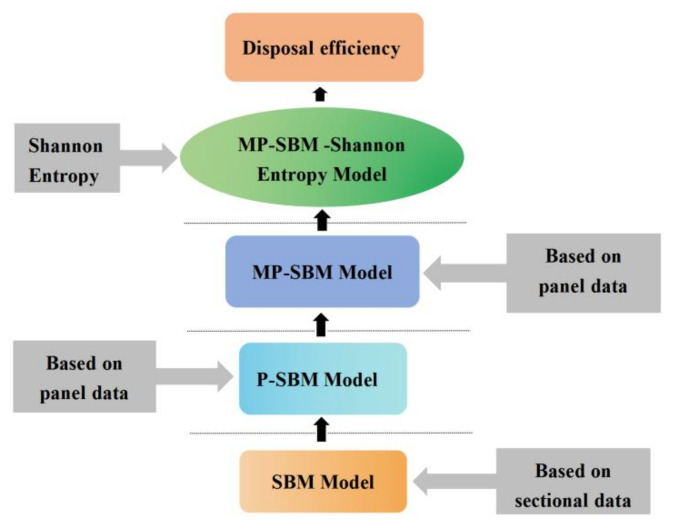
The research framework.

**Figure 2 entropy-24-01608-f002:**
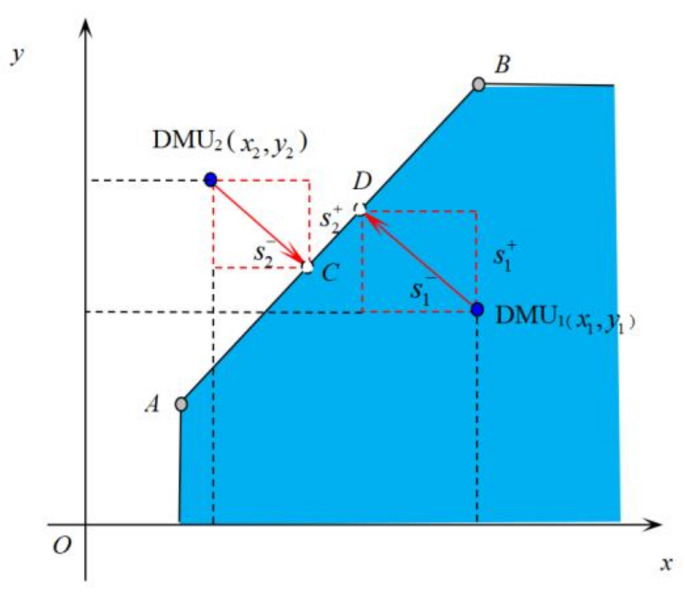
The efficiency measure of the DMU.

**Figure 3 entropy-24-01608-f003:**
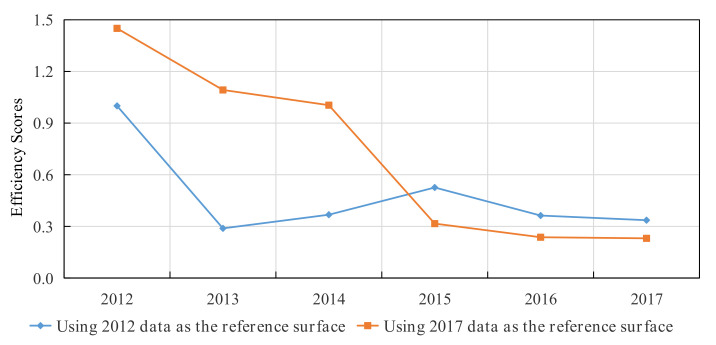
Pure technical efficiency status of Tibet from 2012 to 2017.

**Figure 4 entropy-24-01608-f004:**
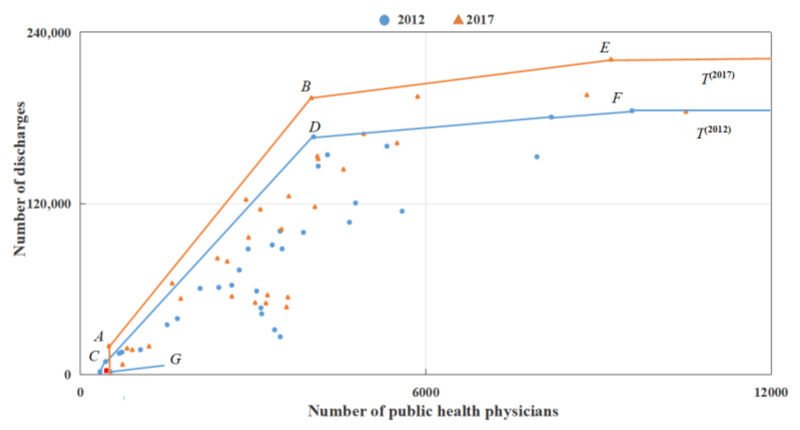
Analysis of the causes of the efficiency paradox of the DMU.

**Figure 5 entropy-24-01608-f005:**
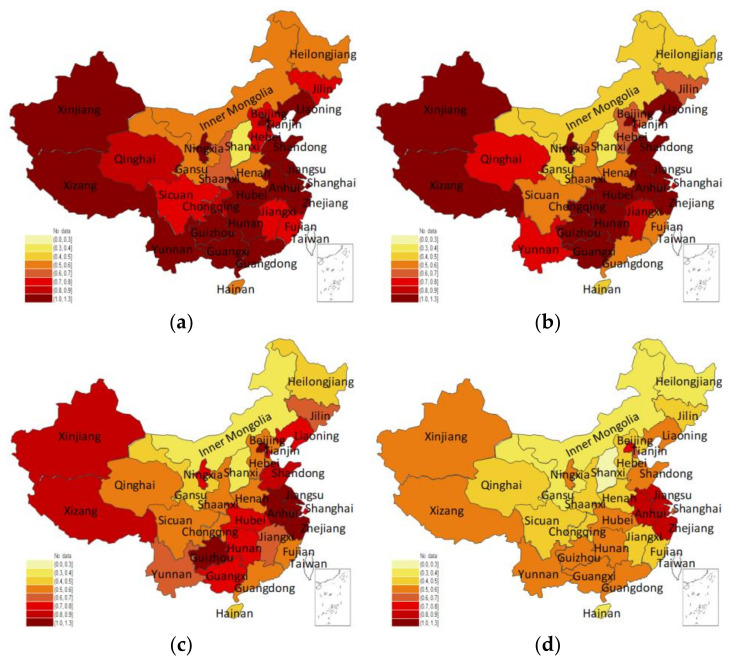
Average provincial disposal efficiency scores under three models. (**a**) Reference surface based on the data of 2012. (**b**) Reference surface based on the data of 2018. (**c**) Reference surface based on the data of 2012–2018. (**d**) The MP-SBM model was applied to measure the efficiency matrix of a subset of 279 variables and combined with the Shannon entropy model.

**Figure 6 entropy-24-01608-f006:**
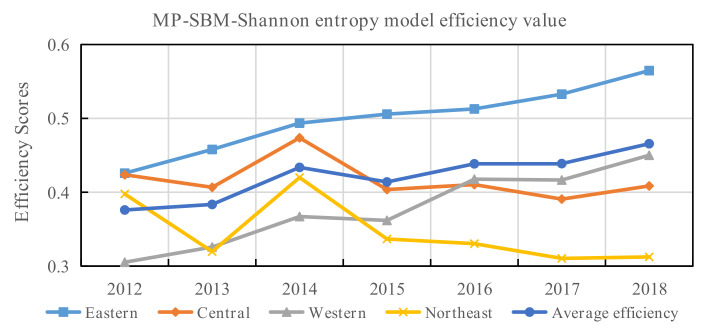
The disposal efficiency trends in China, 2012–2018.

**Figure 7 entropy-24-01608-f007:**
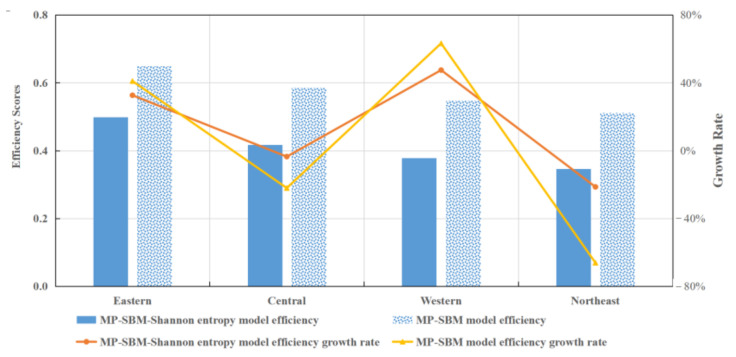
Average regional efficiency and growth rate in China.

**Figure 8 entropy-24-01608-f008:**
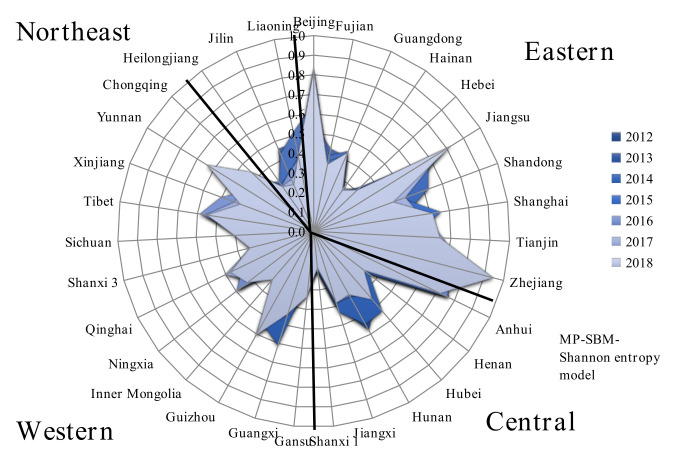
Provincial disposal efficiency, 2012–2018.

**Figure 9 entropy-24-01608-f009:**
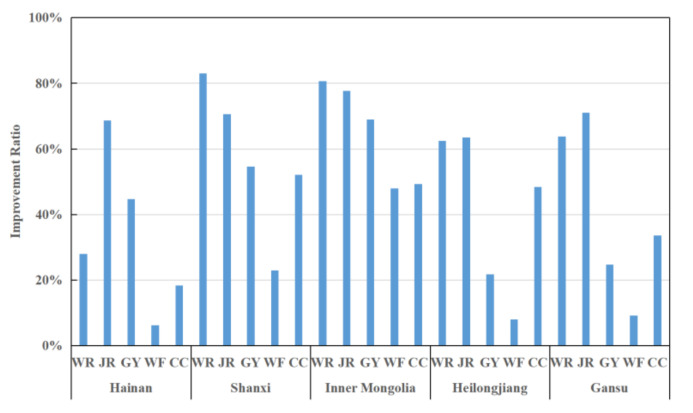
Improvement strategies for input indicators in inefficient provinces.

**Figure 10 entropy-24-01608-f010:**
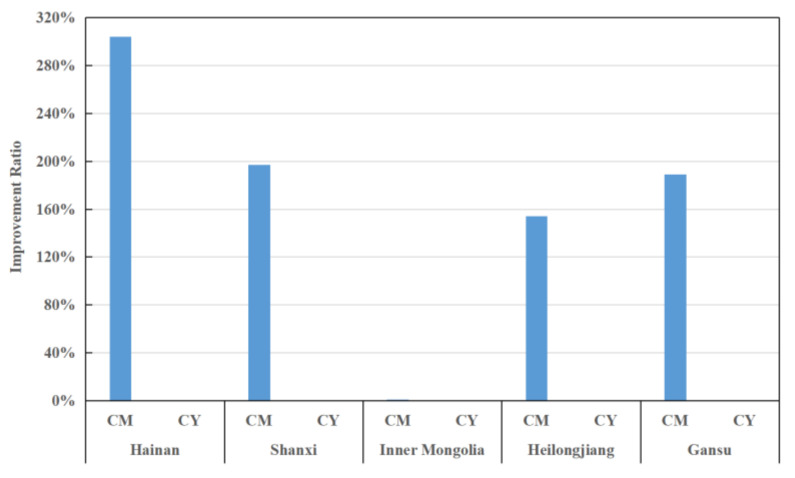
Improvement strategies for desirable output indicators in inefficient provinces.

**Figure 11 entropy-24-01608-f011:**
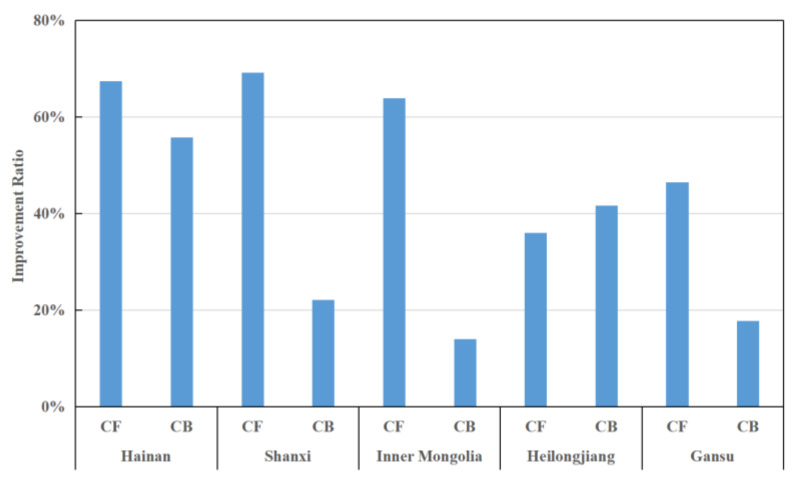
Improvement strategies for undesirable output indicators in inefficient provinces.

**Table 1 entropy-24-01608-t001:** Summary of the related literature indicator system.

Author	Input Indicators	Output Indicators
Zheng et al. [26]	Number of township health centers, number of beds, number of health technicians, number of other personnel	Number of outpatient medical visits, number of hospital discharges, bed occupancy rate, average length of stay in township health centers
Ozcan et al. [27]	Total number of hospital beds, number of people working in related medical departments, number of CT scanners	Number of inpatient discharges, number of outpatient consultations
Rouyendegh et al. [59]	Number of physicians, total number of beds	Bed utilization, total number of procedures, number of patients
Campos et al. [44]	Total healthcare costs, percentage of public investment transferred to labor costs	Number of nursing services for residents, number of specialized healthcare services, number of primary healthcare services
Karahan et al. [45]	Number of beds, total number of doctors, total number of nurses	Number of patients treated, number of inpatients
Vitezić et al. [60]	Employee wages, direct costs, total investment, number of public healthcare departments	Total revenue, number of analyses in various public healthcare sectors
Kawaguchi et al. [61]	Number of hospital managers, maintenance staff, physicians, nurses, nurse assistants, and medical technicians	Medical revenues, inpatient visits, outpatient visits

**Table 2 entropy-24-01608-t002:** An indicator system for evaluating the efficiency of healthcare systems.

Input Indicators	Output Indicators	
	Desirable Output	Undesirable Output
Health surveillance staff (WR), number of CDCs (JR), number of public health physicians (GY), total health expenditure (WF), number of beds in the infectious disease department (CC)	The number of outpatients in DID (CM), the number of discharged patients in DID (CY)	The number of patients in DID (CF), the number of deaths in DID (CB)

**Table 3 entropy-24-01608-t003:** Descriptive statistics of data used in this study (2012–2018, 31 provinces).

Year	Indicator	WR	JR	GY	WF	CC	CM	CY	CF	CB
	Variables	$x_{1}$	$x_{2}$	$x_{3}$	$x_{4}$	$x_{5}$	$y_{1}$	$y_{2}$	$y_{1}^{b}$	$y_{2}^{b}$
2012–2018	Max	7780	17,931	11,175	4622	10,714	561	230,498	387,240	3441
	Min	23	928	342	65	231	2	2223	7113	15
	Mean	2376	6180	3610	1293	3910	125	94,914	99,644	571
	Range	7757	17,003	10,833	4557	10,483	559	228,275	380,127	3426
	Growth rate	−13.5%	−2.8%	5.4%	108.4%	22.8%	64.0%	27.0%	−4.7%	18.4%

**Table 4 entropy-24-01608-t004:** Shannon entropy calculation results.

	$x_{1}$	$x_{2}$	$x_{3}$	$x_{4}$	$x_{5}$	$y_{1}$	$y_{2}$	$y_{1}^{b}$	$y_{2}^{b}$	*W* _k_
1	0	0	0	0	1	1	0	1	0	0.0035991
2	0	0	0	0	1	1	0	1	1	0.0035989
3	0	0	0	0	1	1	0	0	1	0.0035984
4	1	0	0	0	0	1	0	1	0	0.0035982
5	1	0	0	0	0	1	0	1	1	0.0035981
6	1	0	0	0	0	1	0	0	1	0.0035971
7	0	0	0	1	1	1	0	1	0	0.0035969
8	0	0	0	1	1	1	0	1	1	0.0035969
9	0	0	0	1	0	0	1	0	1	0.0035967
10	1	0	0	0	1	1	0	1	1	0.0035965
11	0	1	0	0	1	1	0	1	0	0.0035965
12	0	1	0	0	1	1	0	1	1	0.0035965
13	1	0	0	0	1	1	0	1	0	0.0035965
14	0	0	0	1	1	1	0	0	1	0.0035962
15	0	0	0	1	0	1	0	1	1	0.0035961

**Table 5 entropy-24-01608-t005:** Ranking of disposal efficiency scores based on three models.

Region	Province	P-SBM Model (2012)			MP-SBM Model			MP-SBM-Shannon Entropy Model		
		Efficiency	Average Efficiency	Rank	Efficiency	Average Efficiency	Rank	Efficiency	Average Efficiency	Rank
Eastern	Beijing	1.1653	0.9634	4	1.0000	0.7115	1	0.8256	0.5647	3
	Fujian	0.6002		23	0.4186		19	0.3511		22
	Guangdong	1.1422		7	0.4946		13	0.4404		13
	Hainan	0.4346		30	0.3212		23	0.2556		20
	Hebei	0.5191		27	0.3991		20	0.3078		27
	Jiangsu	1.2161		3	1.0000		1	0.8269		2
	Shandong	1.0417		15	0.6443		3	0.4329		15
	Shanghai	1.1484		6	0.8370		2	0.6147		7
	Tianjin	1.0936		10	1.0000		1	0.6458		5
	Zhejiang	1.2732		2	1.0000		1	0.9461		1
Central	Anhui	1.1653	0.7259	5	1.0000	0.5510	1	0.7196	0.4085	4
	Henan	0.5205		26	0.4270		18	0.3264		25
	Hubei	1.0133		19	0.6329		4	0.4615		11
	Hunan	0.7186		21	0.4934		14	0.3707		21
	Jiangxi	0.6373		22	0.5041		12	0.3844		20
	Shanxi 1	0.3004		31	0.2485		24	0.1885		31
Western	Gansu	0.5683	1.0012	25	0.4337	0.6382	17	0.3372	0.4501	24
	Guangxi	1.0515		14	0.5594		6	0.4435		12
	Guizhou	1.0636		12	1.0000		1	0.6096		8
	Inner Mongolia	1.0186		16	0.4785		15	0.3255		26
	Ningxia	0.7709		20	0.5421		8	0.4288		16
	Qinghai	1.0524		13	0.6091		5	0.4744		10
	Shanxi 3	0.5731		24	0.4398		16	0.3385		23
	Sichuan	1.0771		11	0.5132		11	0.4071		17
	Tibet	1.6153		1	1.0000		1	0.5527		18
	Xinjiang	1.0141		18	0.5318		9	0.3995		9
	Yunnan	1.0944		9	1.0000		1	0.6438		6
	Chongqing	1.1150		8	0.5507		7	0.4401		14
Northeast	Heilongjiang	0.4703	0.6503	28	0.3598	0.4072	21	0.2791	0.3124	28
	Jilin	0.4643		29	0.3338		22	0.2592		29
	Liaoning	1.0163		17	0.5280		10	0.3988		19

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
