# Peer review of "Using Shannon Entropy to Improve the Identification of MP-SBM Models with Undesirable Output"

_entropy, 2022, doi:10.3390/e24111608_

Round 1
Reviewer 1 Report
Attached review report word file

Reviewer 2 Report
The paper entitled “Using Shannon entropy to improve the identification of MP- 2 SBM models with undesirable output” adapts the traditional Slacks-based model to a panel data application using both good and bad outputs. The undesirable (bad) outputs are considered in the original efficiency matrix combination under Shannon’s entropy perspective to cope with existing ranking limitations in the DEA model. I believe the paper has potential, but some aspects must be highlighted or corrected. The work lacks motivation and literature-based justification in many parts. Please address my comments ahead.
1. In the abstract, the authors mention an “efficiency paradox” which I have never heard about it. They state that the traditional SBM model has this paradox, which is just not true. From what I have read in your manuscript (subsections 2.3.1 and 2.3.2), this is a particular empirical situation in your data analysis that may happen when you construct the production frontier using panel data for the linear combination regarding a different reference set. This is not an issue with the model, you should not generalize it, otherwise you would be assuming that every other DEA-SBM application (and there are thounsands of them) are potentially wrong. In fact, the following explanation for the so-called paradox “the higher the evaluation criteria, the greater the efficiency value for the same decision-making unit(DMU)” makes it even more confusing for the reader. I suggest properly explaining it in the abstract or simply removing it from the abstract and mentioning that a particular efficiency paradox will be discussed.
2. The paper lacks reference and literature foundation in many parts. Please reference the efficiency paradox (line 39-40) and comprehensive efficiency score(CES) (line 58). In addition, some papers that can be interesting to cite in the introduction when discussing the many applications and different DEA models are reviews in the field. Some suggestions are the reviews of Daraio et al., Liu, Cook and Seiford:
· Daraio et al. (2020). Empirical surveys of frontier applications: a meta‐review. International Transactions in Operational Research, 27(2), 709-738.
· Daraio et al. (2019). Productivity and efficiency analysis software: an exploratory bibliographical survey of the options. Journal of Economic Surveys, 33(1), 85-100.
· Cook, W. D., & Seiford, L. M. (2009). Data envelopment analysis (DEA)–Thirty years on. European journal of operational research, 192(1), 1-17.
· Liu, J. S., Lu, L. Y., Lu, W. M., & Lin, B. J. (2013). Data envelopment analysis 1978–2010: A citation-based literature survey. Omega, 41(1), 3-15.4
· Seiford, L. M. (1996). Data envelopment analysis: the evolution of the state of the art (1978–1995). Journal of productivity analysis, 7(2), 99-137.
3. In addition, the context of application is public health, nevertheless, there is no literature section discussing the relevant DEA contributions in this field, with a particular focus on China, and I know there is plenty work on this. Some other suggestions of important developments and reviews to reference SBM applications in hospitals are:
· Azreena, E., Juni, M. H., & Rosliza, A. M. (2018). A systematic review of hospital inputs and outputs in measuring technical efficiency using data envelopment analysis. International Journal of Public Health and Clinical Sciences, 5(1), 17-35.
· Gok, M. S., & Sezen, B. (2011). Analyzing the efficiencies of hospitals: An application of Data Envelopment Analysis. Journal of Global Strategic Management, 10(1), 137-146.
· Kohl, S., Schoenfelder, J., Fügener, A., & Brunner, J. O. (2019). The use of Data Envelopment Analysis (DEA) in healthcare with a focus on hospitals. Health care management science, 22(2), 245-286.
· Nepomuceno et al. (2020). A DEA-based complexity of needs approach for hospital beds evacuation during the COVID-19 outbreak. Journal of healthcare engineering.
· Nepomuceno et al. (2022). The Core of Healthcare Efficiency: A Comprehensive Bibliometric Review on Frontier Analysis of Hospitals. In Healthcare (Vol. 10, No. 7, p. 1316). MDPI.
4. Other suggestions for enriching references in Banks (line 35):
· An, Q., Liu, X., Li, Y., & Xiong, B. (2019). Resource planning of Chinese commercial banking systems using two-stage inverse data envelopment analysis with undesirable outputs. Plos one, 14(6), e0218214.
· Kaffash, S., & Marra, M. (2017). Data envelopment analysis in financial services: a citations network analysis of banks, insurance companies and money market funds. Annals of Operations Research, 253(1), 307-344.
· Nepomuceno, T. C., & Costa, A. P. C. (2019). Resource allocation with time series DEA applied to Brazilian federal saving banks. Economics Bulletin, 39(2), 1384-1392.
5. and educational institutions (line 36):
· Johnes, J. (2006). Data envelopment analysis and its application to the measurement of efficiency in higher education. Economics of education review, 25(3), 273-288.
· Kuah, C. T., & Wong, K. Y. (2011). Efficiency assessment of universities through data envelopment analysis. Procedia computer science, 3, 499-506.
· Thanassoulis, E., Witte, K. D., Johnes, J., Johnes, G., Karagiannis, G., & Portela, C. S. (2016). Applications of data envelopment analysis in education. In Data envelopment analysis (pp. 367-438). Springer, Boston, MA.
6. Please put a space in between words and the “(“ : decision-making unit(DMU), P-SBM(Panel Slacks-based Measure), model(Modified Panel Slacks-based Measure Shannon entropy model), comprehensive efficiency score(CES) ….
7. Please define “CDC” (line 83) and “DID” (line 86).
8. I believe justifications (i.e. “avoid efficiency paradox – figure 1) are not part of a framework representation. It should illustrate the systematic methodological steps of your analysis. This is just a minor comment. You may consider adjusting it.
9. The first unit invariant slacks-based measure was actually proposed by Charnes et al. (1985) “Foundations of data envelopment analysis and Pareto–Koopmans empirical production functions. Journal of Econometrics 30, 91–107”. This was proposed to cope with Russell (1988) non-commensurate measures of technical efficiency. It was Green et al. (1997) “A note on the additive data envelopment analysis model. Journal of the Operational Research
Society 48 (4), 446–448” that proposed the [1/(s+m)]*(…) functional that you formalize in your paper subjected to the same constraints, and 1- [1/(s+m)]*(…) as a legit unit invariant efficiency score. Tone (2001) SBM was based on those previous approaches that should be mentioned in the text.
10. Please define all variables and indexes (i, t, r, lambda, b, n s-, s+, k0, l…) m the equations 1, 2 and 3.
11. Line 112: comprehensive efficiency score. What do the authors mean by that? Is it the technical efficiency (TE) under CRS?
12. Line 115: Technical efficiency values. This is in fact the pure technical efficiency (PTE) under VRS technology. The technical efficiency is PTE*SE (scale efficiency).
13. Lines 118 – 120. Strong assumption without any reference. It generalizes a problem that I can only see in your application, as I mentioned before in the first comment. I suggest you remove it or provide a strong foundation for this.
14. Model (2) is basically Tone’s SBM applied to panel data under different reference sets. It is unit invariant and monotone increasing on each input/output slack configuration. y1 is in the unit scale (0 ≤ y1 ≤ 1). I cannot see how it would be more than 1 (line 124). I cannot see this as a valid justification for using model (3) (which has an inverted frontier perspective). Please explain it, or provide a solid justification for model 3.
15. I believe the equation on line 133 is unnecessary since you have formalized all the linear models. If the authors want to keep it, please number the equation as well.
16. I believe figure 2 refers to the model of equation (3). Is it so?
17. Are you considering data on Tibet healthcare institutions from 2012 to 2014 (lines 163 and 168) or from 2012 to 2017 (line 173)?
18. The figures representing Jiangxi and Ningxia are redundant without a discussion. You have made your point already in figure 3. There is no need to report other regions if there is no additional information, only figures. Please comment more about the illustrations or remove them.
19. I am having problems accepting this paradox, and this is my most important comment. From my perspective, the fact that Tibet’s technical efficiency does not decrease with the increase in China’s technological progress could be explained by so many exogenous or endogenous factors that are not included in your representations. It is not necessarily a data paradox. The analysis in 2.3 is too simple to generalize this assumption. It could change by including or removing different healthcare resources or products or by controlling exogenous determinants of technological change. Another thing is that the panel DEA analysis used to report this issue is not in the standard way that I am used to seeing. I am more familiar with the panel representation of subsection 2.4, equation 6. Why do the authors use a different reference set (China) to construct the non-parametric production frontier that Tibet is compared to? There is no problem with that, but this is not usual. Authors should justify this approach and be honest in recognizing that your so-called paradox is a problem specific to the context and methodology that you are applying.
20. Last but not least, the main reason I accepted reviewing this manuscript is that it combines my research field (non-parametric frontier estimations) with a topic that I am interested in but still have limited knowledge (entropy). Unfortunately my limited knowledge of the properties and concepts of Shannon's Information Entropy did not change a bit by reading the paper. It lacks fundamental concepts and entropy details for the journal audience, and a link with the healthcare context that can be provided with additional literature reviews.
I believe the paper has plenty of potentials and I encourage major reviews addressing all the comments from above prior to a publication.

Round 2
Reviewer 1 Report
Pls explain Super-SBM-DEA model advantage.
Reviewer 2 Report
I believe the authors have addressed my concerns.
Author Response
We would like to thank you for your time spent in reading and processing our manuscript, and providing constructive comments to help us improve the paper.